

# High-frequency repetitive transcranial magnetic stimulation protects against 6-OHDA-induced Parkinson's disease symptoms by modulating the proNGF-p75NTR-sortilin pathway

Rui Zhao[1,2,3,*], Wanqing Du[1,2,4,*], Ke Tian[1,3], Kunlong Zhang[1], Hua Yuan[1], Fang Gao[2], Xin Kang[1,2] and Xiaolong Sun[1]

[1] Department of Rehabilitation Medicine, Xijing Hospital, Fouth Military Medical University, Xi'an, China
[2] Department of Neurobiology, Institute of Neurosciences, Fouth Military Medical University, Xi'an, China
[3] College of Life Sciences, Northwest University, Xi'an, China
[4] Department of Orthopedics, The Second Affiliated Hospital of Xi'an Medical University, Xi'an, China
[*] These authors contributed equally to this work.

Corresponding authors
Xin Kang, kangxin0928@foxmail.com
Xiaolong Sun, xlsun@fmmu.edu.cn

## ABSTRACT

**Background**. High-frequency repetitive transcranial magnetic stimulation (HF rTMS) is a promising non-invasive treatment for Parkinson's disease (PD) in clinical settings. However, the precise mechanisms are incompletely understood. The proNGF-p75NTR-sortilin signaling pathway is closely associated with nigrostriatal dopaminergic (DA) neuron degeneration, a hallmark feature of PD. This study aimed to evaluate the neuroprotective efficacy of HF rTMS on the proNGF-p75NTR-sortilin pathway in PD.

**Methods**. Using a PD rat model induced by 6-hydroxydopamine (6-OHDA), the rats were randomly divided into two groups: sham rTMS group and HF rTMS group. After a 4 w intervention, the rats' motor function was assessed using a rotarod test. Additionally, immunofluorescence technology was used to analyze the DA neurons in the midbrain of PD rats, and immunohistochemistry and Western blot analysis were employed to evaluate the expression levels and effects of proNGF-p75NTR-sortilin in the midbrain following HF rTMS.

**Results**. Our research revealed a significant increase of proNGF expression in reactive astrocytes and upregulated sortilin and p75NTR in DA neurons in the substantia nigra of the hemisphere ipsilateral to the induced lesion, correlated with the degeneration of DA neurons and impaired motor functions. A four-week regimen of HF rTMS, as opposed to sham rTMS, significantly improved the above pathological conditions. The decrease in proNGF-p75NTR-sortilin expression following HF rTMS correlated with a significant increase in DA neuron survival and motor function improvement. HF rTMS exhibited no effects on proBDNF expression. Our study findings indicate that the targeted proNGF-p75NTR-sortilin complex modulation may be an essential mechanism through which HF rTMS exerts its neuroprotective effect. This insight offers a new therapeutic perspective for PD management, highlighting the potential of rTMS to modulate key neurodegenerative pathways.

# INTRODUCTION

Parkinson's disease (PD) is the second most common neurodegenerative disorder globally (*Fahn, 2018*). The characteristic clinical features of PD include motor disturbances such as muscle rigidity, resting tremors, and bradykinesia (*Gandhi & Wood, 2005*; *Santa-Cecilia et al., 2019*). PD pathogenesis is based on the progressive degeneration and death of dopaminergic (DA) neurons within the substantia nigra (SN), and the underlying causes of this selective neuronal loss are unknown (*Nemade, Subramanian & Shivkumar, 2021*). Pharmacological approaches and surgical interventions are both associated with a variety of side effects (*Olson et al., 2023*). Long-term use of levodopa may trigger adverse reactions such as dyskinesia and involuntary movements (*Bastide et al., 2015*). Although deep brain stimulation (DBS) is currently regarded as an effective therapeutic approach, the high cost of surgery and the risk of postoperative infection should not be overlooked either (*Umemura et al., 2016*). In the modern clinical framework, there is an increasing interest in non-invasive stimulation methods and neurotrophic factor-based therapies (*Church, 2021*; *Mally et al., 2018*). However, the empirical evidence supporting the efficacy of these innovative treatments remains relatively rare.

Repetitive transcranial magnetic stimulation (rTMS) is a non-invasive neuromodulation method that demonstrates potential therapeutic applications in various neurological and psychiatric conditions (*Lefaucheur et al., 2020*; *Iglesias, 2020*). rTMS has demonstrated multifaceted potential therapeutic effects in the treatment of PD. A randomized, double-blind, controlled study showed that after 2 w of HF rTMS (10 Hz) treatment, consisting of 10 sessions, patients experienced a significant reduction in freezing of gait (FOG) scores, along with significant improvements in gait parameters such as walking speed and step frequency (*Mi et al., 2020*). Empirical research has demonstrated that long-term rTMS can induce neuroprotective effects and enhance motor deficits in a rat model of PD induced by 6-hydroxydopamine (6-OHDA) (*Hsieh et al., 2021*). The therapeutic efficacy of rTMS depends on the stimulation frequency. A previous study reported that high-frequency (HF) and low-frequency (LF) rTMS can exert beneficial effects on motor function (*Lee et al., 2015*). Our research group investigated this phenomenon and reported that HF rTMS significantly reduced post-Parkinsonian astrocyte activation by modulating the endocannabinoid and endocannabinoid receptor 2 pathways, an effect not observed with LF rTMS (*Kang et al., 2022*). These findings indicate that HF rTMS may provide superior therapeutic benefits in PD treatment.

Neurotrophic factors are essential in the health and disease of the nervous system, especially in neurodegenerative diseases, including PD (*Bruno et al., 2023*). These factors bind to specific receptors and promote neuronal survival, proliferation, maturation, and maintenance of function (*Sampaio et al., 2017*). The nerve growth factor (NGF) is among the earliest identified neurotrophic factors and is crucial for the survival of sensory and

sympathetic neurons (*Fahnestock & Shekari, 2019*). In the context of PD, NGF activity may be regulated, affecting DA neurons' stability and functions (*Xia et al., 2013*; *Wang et al., 2008*). Brain-derived neurotrophic factor (BDNF) is another essential neurotrophic factor that promotes neuronal survival and synaptic plasticity (*Wang et al., 2024*; *Azman & Zakaria, 2022*). As precursor forms of NGF and BDNF, proNGF and proBDNF exert different biological effects from their mature forms (*Bradshaw et al., 2015*; *Jia et al., 2023*). They can induce neuronal apoptosis by binding to p75NTR and sortilin receptors. Sortilin is a multifunctional receptor involved in the intracellular trafficking and sorting of various ligands, including neurotrophic factors and pro-inflammatory cytokines. Its role in regulating these pathways is crucial for maintaining cellular homeostasis and has been implicated in several neurodegenerative processes (*Chen et al., 2008*). The proNGF-sortilin signaling complex has the capacity to induce neuronal apoptosis, a form of programmed cell death. This mechanism is implicated in the degeneration and demise of DA neurons, thereby fueling the progression of PD. In stark contrast, the proNGF-p75NTR-sortilin complex has been repeatedly proven to be a viable target for protecting SN neurons in PD treatment. It notably plays a crucial role in regulating the delicate balance between neuronal death and survival (*Meek et al., 2024*; *Dechant & Barde, 2002*). Modulating the activity of the proNGF-p75NTR-sortilin complex may hold the key to reducing neuronal apoptosis while promoting neuronal survival and functional recovery, thus presenting a new avenue for PD treatment. This study aimed to determine whether HF rTMS may exert neuroprotective effects by modulating the proNGF-p75NTR-sortilin pathway in PD.

## MATERIALS AND METHODS

### Rat

Male Sprague–Dawley rats (6 weeks, 180–200 g) were procured from the Animal Centre of the Fourth Military Medical University in China, which were randomly divided into Sham rTMS (20 rats)and HF rTMS (20 rats) groups. These animals were raised in an animal facility with three to four animals per cage, a constant temperature of 22 °C, regulated temperature of 25 °C within a 12-h light/dark cycle with ad libitum access to adequate food and water. They were acclimated for at least 10 days before surgery was performed. All procedures were conducted with the highest regard for minimizing animal usage and mitigating potential distress. All animal experiments were approved by the Committee of Animal Use for Research and Education of FMMU (Approval date: 30/06/2023; Approval No. IACUC-20230558). and were conducted in accordance with the National Institutes of Health guide for the care and use of laboratory animals (NIH Publications No. 80–23, revised 1996).

### Preparation of the 6-OHDA models

The 6-OHDA lesion model was established following protocols from our previous research, with minor adjustments (*Kang et al., 2022*). The rats were anesthetized and administered unilateral injections of 40 µg 6-OHDA (20 µg/4 µL/site; Sigma) into the left striatum at two separate regions. The stereotaxic coordinates for the injections were as follows: anterior-posterior (AP) + 0.5 mm, mediolateral (ML) + 2.5 mm, dorsoventral (DV)

− 5.0 mm; and AP − 0.5 mm, ML + 4.2 mm, DV − 5.0 mm. The 6-OHDA was formulated by diluting it in a solution of 0.9% saline and 0.02% ascorbic acid. Equal volumes of 0.9% saline were administered as a control into the corresponding sites on the right hemisphere of the brain (*Kang et al., 2022*).

## Apomorphine (APO)-induced rotation test

One week post-6-OHDA injection, the rats received apomorphine (WAKO, 0.5 mg/kg, intraperitoneally) to induce rotational behavior, a standard assay for assessing PD model efficacy (*Su et al., 2018*). Initially, the rats were placed in a cylindrical container and allowed to acclimate for 30 min. After the drug administration, an anonymized observer recorded the number of rotations performed by each rat for 30 min, commencing 5 min after the injection. Rats that exhibited >60 contralateral rotations within a 30-min window were regarded as successful PD models and were subsequently included in the subsequent experimental procedures. Behavioral assessments were repeated 4 w post-inoculation to assess the progression and response to treatment interventions.

## rTMS

The PD rats were divided into two distinct groups: the sham rTMS and HF rTMS groups. The rTMS protocol was based on that of a previous study (*Kang et al., 2022*), incorporating the following parameters: 500 pulses daily, 8 min and 20 s stimulation duration, and a pulse intensity set at 20% of the rTMS device's maximum output. For the HF rTMS group, the center of the coil was positioned directly above the apex of the skull during the stimulation sessions. In the sham rTMS group, the coil was placed 10 cm above the head to simulate the sensation of coil vibration without delivering actual brain stimulation, allowing the rats to perceive tactile feedback devoid of magnetic field influences.

The animals were subjected to a 4 d acclimatization period to minimize the impact of stress on the experimental outcomes. This included daily 10 min sham stimulation sessions to familiarize them with the procedure. During stimulation, the rats were carefully restrained by hand to ensure consistency and control over potential stressors. This careful preparation aimed to standardize the experimental conditions and ensure that any observed effects could be attributed to the treatment rather than procedural stress.

One week after the 6-OHDA injection, the rats received apomorphine to induce rotational behavior and subsequently underwent rTMS intervention. After a 4-week course of magnetic stimulation, the rats underwent Western blotting (WB) analysis, immunofluorescence staining, and behavioral assessments. In the sham rTMS group, a simulated stimulation procedure was conducted without actual magnetic stimulation applied (Fig. 1A).

## Western blotting

The rats (sham rTMS $n = 6$, HF rTMS $n = 6$) were humanely euthanized by decapitation, and their brains were promptly extracted and stored on ice within brain molds. Tissue samples from the ventral midbrain were carefully dissected and homogenized in RIPA lysis buffer, supplemented with phosphatase and EDTA inhibitors. The homogenate was centrifuged at 12,000 rpm for 5 min to remove sedimented debris. The protein

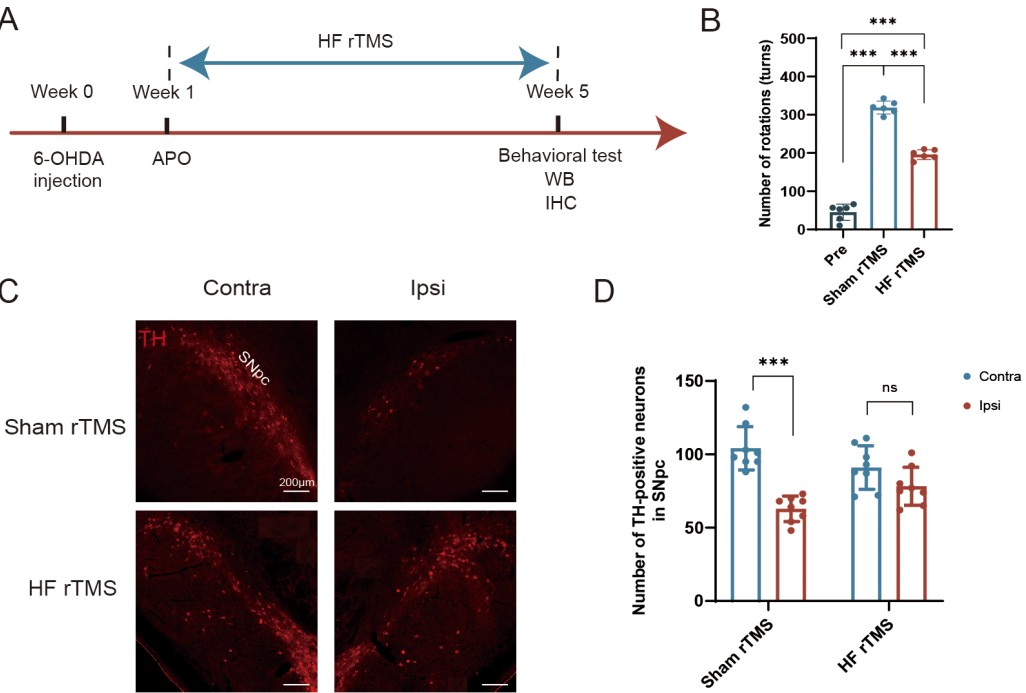

**Figure 1** **Protective effect of HF rTMS on DA neurons in a 6-OHDA rat model.** (A) Schematic diagram of the 6-OHDA PD model and HF rTMS. (B) Quantification of the apomorphine-induced rotations in Pre, sham rTMS and HF rTMS groups by one-way ANOVA with Bonferroni's multiple comparison test. (C) Representative immunofluorescence demonstrating TH-positive neurons in the contralateral or ipsilateral SNpc at sham rTMS and HF rTMS groups. Bar = 200 μm. (D) Quantification of the numbers of TH-positive neurons in SNpc by paired $t$-test. Data are expressed as mean ± SD ($n = 6$–8/group) (***$P$ < 0.001).

concentration in the supernatant was quantified using the bicinchoninic acid assay, enabling the normalization of protein content for subsequent analyses. The protein extracts were separated using on-gel electrophoresis on a 10% sodium dodecyl sulfate-polyacrylamide gel (SDS-PAGE) at a constant voltage of 120 V. Following electrophoresis, the resolved proteins were transferred to a polyvinylidene difluoride membrane at 100 V. The membranes were subsequently blocked for 2 h to inhibit non-specific binding. After blocking, the membranes were incubated overnight at 4 °C with primary antibodies targeting proNGF (1:1000; Abcam), p75NTR (1:1000; CST), proBDNF (1:1000; Abcam), and sortilin (1:1000; Abcam). Subsequently, incubation was performed with horseradish peroxidase-conjugated secondary antibodies (1:4000) procured from Cell Signaling Technology (Boston, MA, USA). The immunoblots were developed using an enhanced chemiluminescence detection system and subsequently exposed to radiographic film. The immunoreactive bands on the blots were quantified using ImageJ software, and the optical densities were normalized to the housekeeping protein GAPDH to control for variability in protein loading. This normalization step ensured the accuracy and reliability of the comparative analysis of protein expression levels across different experimental conditions.

## Immunofluorescent labeling

The rats from the sham rTMS ($n = 6$) and HF rTMS groups ($n = 6$) were anesthetized with a 1% sodium pentobarbital solution and subsequently perfused with saline through the heart, followed by perfusion with 4% paraformaldehyde (PFA) for tissue fixation. The brains were carefully extracted and postfixed in 4% PFA for 4 h, immersed in a 30% sucrose solution at 4 °C until the tissues became buoyant, signifying sufficient infiltration of the cryoprotectant. Coronal sections of the brain (30 μm in thickness) were prepared using a cryostat. The sections were initially incubated with a blocking solution containing 0.1% Triton X-100 and 5% bovine serum albumin (BSA) for 1 h at room temperature to reduce non-specific binding and permeabilize the cell membranes. The midbrain sections containing the nigra were incubated in the following primary antibodies: mouse anti-tyrosine hydroxylase (TH) (1:30000; Sigma), rabbit anti-proNGF (1:200, Abcam), rabbit anti-p75NTR (1:2000; CST), rabbit anti-sortilin (1:1000; Abcam), and mouse anti-glial fibrillary acidic protein (GFAP) (1:1000; Abcam).This incubation was performed at 4 °C for 24 h to allow for optimal antibody binding. After the primary antibody incubation, the sections were subsequently incubated with the appropriate secondary antibodies: Alexa Fluor 594-conjugated goat anti-mouse IgG (1:800; Abcam) and Alexa Fluor 488-conjugated goat anti-rabbit IgG (1:800; Abcam) for 4 h at room temperature. These secondary antibodies, with their distinct fluorophores, enable the visualization of the various antigens under a fluorescence microscope. Finally, ImageJ was used to assess the number of positive cells and the fluorescence intensity in the brain sections.

## Data analysis and statistics

GraphPad Prism software (version 9.0; GraphPad Software, Inc.) was used for all statistical analyses, and results are expressed as the mean $\pm$ SD ($n = 3$–6) of at least three animal samples or three independent experiments. The comparisons between groups were determined by paired $t$-test or one-way analysis of variance (ANOVA) with Bonferroni's multiple comparison test. Results with a $P < 0.05$ were statistically significant (*$P < 0.05$, **$P < 0.01$, and ***$P < 0.001$).

# RESULTS

## Protective effect of HF rTMS on DA neurons and motor function in 6-OHDA PD model

We performed a comparative analysis of the effects of 6-OHDA lesioning and HF rTMS, following a meticulous experimental schedule (Fig. 1A). The increase in APO-induced rotations observed at five weeks post-6-OHDA administration in the sham rTMS group was significantly reduced in rats that underwent HF rTMS treatment ($P < 0.001$, Fig. 1B). However, there was still a difference compared to pre group ($P < 0.001$, Fig. 1B). The quantity of tyrosine hydroxylase (TH)-positive DA neurons in the ipsilateral SN pars compacta (SNpc) was significantly diminished in the sham rTMS group (Fig. 1C). A comprehensive somatotopic analysis revealed that HF rTMS significantly reduced the 6-OHDA-induced loss of TH-positive neurons ($P < 0.001$, Fig. 1D).

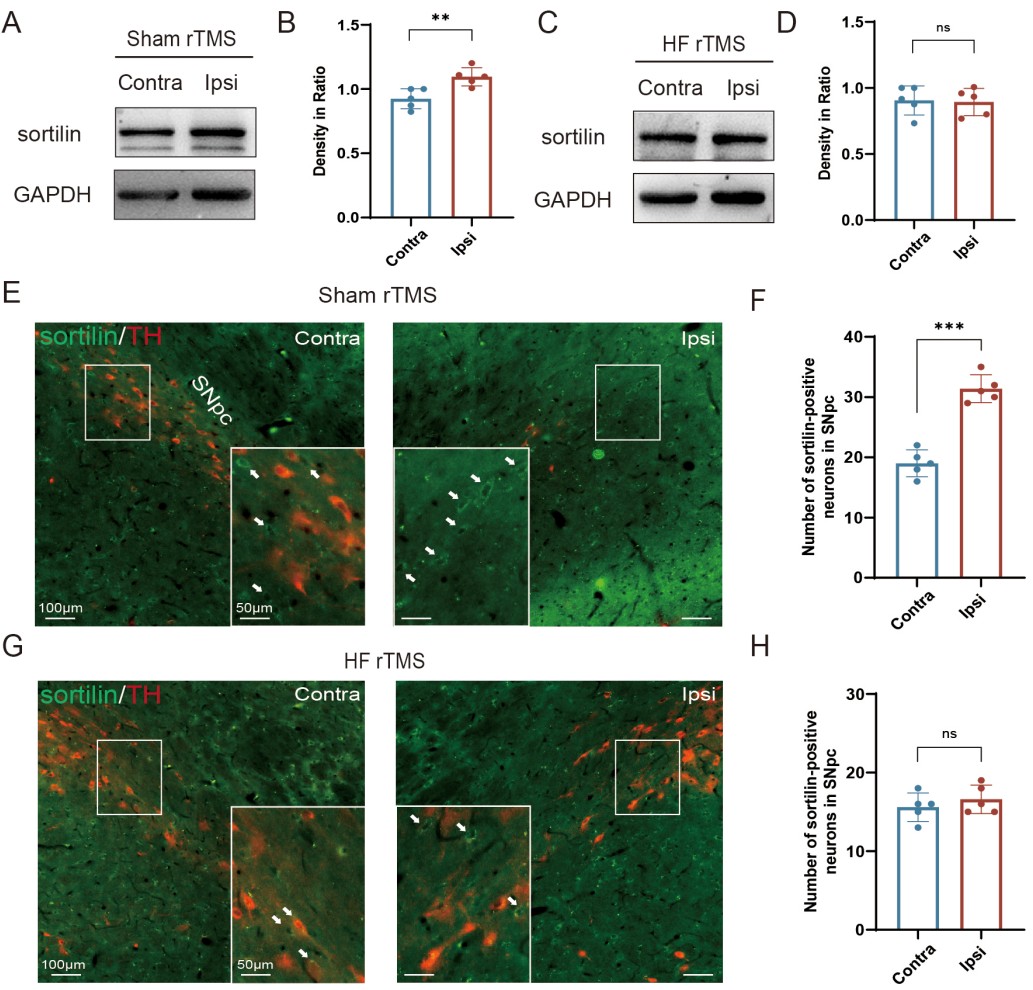

**Figure 2** **HF rTMS attenuated sortilin expression in DA neurons in 6-OHDA-lesioned rat models.**
(A) Representative western blot of sortilin level in the ipsilateral or contralateral ventral midbrain in sham rTMS. (B) Quantitative summary levels of sortilin expression. (C) Representative western blot of sortilin level in the ipsilateral or contralateral ventral midbrain in HF rTMS. (D) Quantitative summary levels of sortilin expression. (E) Representative immunofluorescence demonstrating TH and sortilin-positive neurons in the contralateral or ipsilateral SNpc in sham rTMS. (F) Quantitative summary levels of sortilin-positive neurons in the contralateral or ipsilateral SNpc in sham rTMS. (G) Representative immunofluorescence demonstrating TH and sortilin-positive neurons in the contralateral or ipsilateral SNpc in HF rTMS. (H) Quantitative summary levels of sortilin-positive neurons in the contralateral or ipsilateral SNpc in HF rTMS. Data are expressed as mean ± SD ($n = 5$/group) (***$P < 0.001$ and **$P < 0.01$, by paired $t$-test).

## HF rTMS attenuated sortilin expression in DA neurons in 6-OHDA-lesioned rat models

In the sham rTMS group, western blotting analysis of the midbrain SN tissue lysates revealed a significant increase in sortilin protein levels in the ipsilateral SN ($P < 0.01$ Figs. 2A–2B). HF rTMS treatment significantly reduced sortilin expression, bringing it closer to the levels observed in the contralateral, non-lesioned side (Figs. 2C–2D). Brain sections from the sham rTMS rats underwent double immunofluorescence labeling with

antibodies against sortilin and TH. The resultant data confirmed the protein blotting findings, indicating a significant reduction in ipsilateral TH-positive DA neurons in sham rTMS (Fig. 2E). Additionally, the number of neurons positive with sortilin expression significantly increased ($P < 0.001$, Fig. 2F). After a sustained four-week period of HF rTMS, a significant decrease in ipsilateral sortilin fluorescence intensity was observed, with an increase in TH-positive neurons (Figs. 2G–2H) consistent with the western blotting results.

### HF rTMS down-regulated the expression of p75NTR in TH-positive neurons in 6-OHDA model rats

Correspondingly, we observed a significant upregulation of p75NTR expression on the injured hemisphere of the brain in the sham rTMS group ($P < 0.05$, Figs. 3A–3B). After 4 w of HF rTMS treatment, the aberrant p75NTR upregulation in the ventral region of the injured midbrain was significantly inhibited, demonstrating a non-significant difference from that of the contralateral side (Figs. 3C–3D). Immuno-fluorescent staining confirmed that TH/p75NTR double immunofluorescence further confirmed the ipsilateral upregulation of neuronal p75NTR and decrease of TH-positive DA neurons ($P < 0.01$, Figs. 3E–3F) , and HF rTMS treatment inhibited the up-regulation of p75NTR expression and increased DA neuron survival in the 6-OHDA rat model (Figs. 3G–3H).

### HF rTMS reduced the expression of proNGF in astrocytes without affecting the expression of proBDNF

In the sham rTMS group, proNGF expression was significantly increased in the injured ventral midbrain compared to the contralateral side ($P < 0.05$, Figs. 4A–4B). After HF rTMS treatments, proNGF expression was significantly diminished in the ventral midbrain region surrounding the injury (Figs. 4C–4D). Double immunofluorescence staining confirmed that the expression of astrocytes on the affected side increased and the expression of proNGF in them was also significantly upregulated ($P < 0.05$, Figs. 4E–4F). Additionally, We observed reduced proNGF expression in astrocytes after 4 weeks of HF rTMS (Figs. 4G–4H). Similar to proNGF, we found a significant increase in proBDNF levels on the ipsilateral side ($P < 0.05$, Figs. 4I–4J). However, the HF rTMS application did not significantly change proBDNF expression ($P < 0.05$, Figs. 4K–4L).

## DISCUSSION

Herein, we observed significant cell death in ipsilateral DA neurons and rotational behavior after 6-OHDA-induced injury, mitigated by HF rTMS treatment. The potential underlying mechanism may involve the downregulated expression of proNGF in reactive astrocytes, p75NTR expression, and sortilin in DA neurons.

A previous clinical study reported that rTMS potentially enhances motor symptoms for patients with PD (Li, Wu & Yi, 2015). Based on the applied frequency, rTMS is clinically classified into HF and LF rTMS, which elicit varying modulatory effects. Animal models have demonstrated that HF (Lee et al., 2013) and LF rTMS (Yang, Song & Liu, 2010; Ba et al., 2017) enhance the survival of DA neurons and improve motor functions, respectively, in

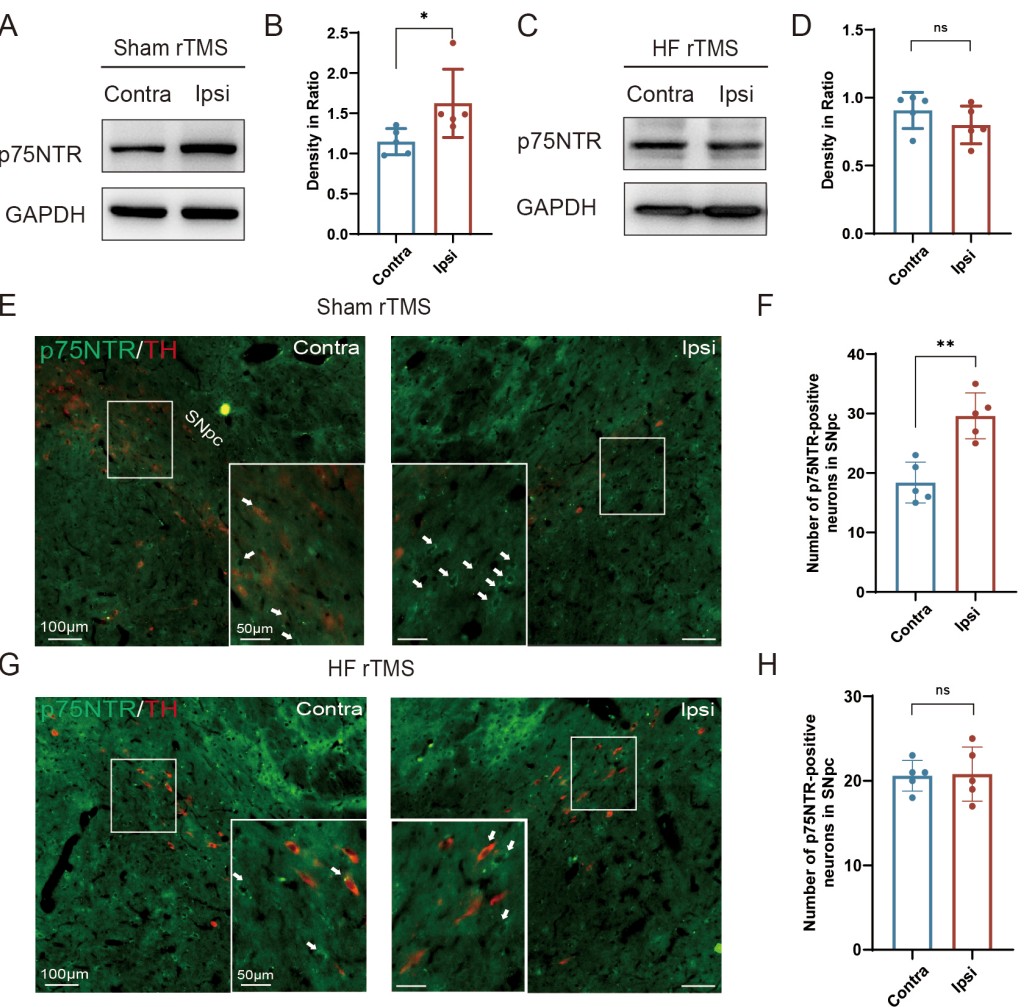

**Figure 3** **HF rTMS down-regulated the expression of p75NTR in TH-positive neurons in 6-OHDA model rats.** (A) Representative western blot of p75NTR level in the ipsilateral or contralateral ventral midbrain in sham rTMS. (B) Quantitative summary levels of p75NTR expression. (C) Representative western blot of p75NTR level in the ipsilateral or contralateral ventral midbrain in HF rTMS. (D) Quantitative summary levels of p75NTR expression. (E) Representative immunofluorescence demonstrating TH and p75NTR-positive neurons in the contralateral or ipsilateral SNpc in sham rTMS. (F) Quantitative summary levels of p75NTR-positive neurons in the contralateral or ipsilateral SNpc in sham rTMS. (G) Representative immunofluorescence demonstrating TH and p75NTR-positive neurons in the contralateral or ipsilateral SNpc in HF rTMS. (H) Quantitative summary levels of p75NTR-positive neurons in the contralateral or ipsilateral SNpc in HF rTMS. Data are expressed as mean $\pm$ SD ($n = 5$/group) (**$P < 0.01$ and *$P < 0.05$ by paired $t$-test).

PD. Clinical studies consistently confirmed that HF and LF rTMS can effectively improve motor function in patients with PD, with HF rTMS treatment potentially exhibiting a more robust and enduring effect than LF rTMS treatment (*Zhu et al., 2015*). Our previous research directly compared the impact of HF and LF rTMS on the survival of DA neurons in a 6-OHDA rat model, revealing that high frequency exhibited better effects than low frequency (*Kang et al., 2022*). This study further confirmed the effectiveness of high

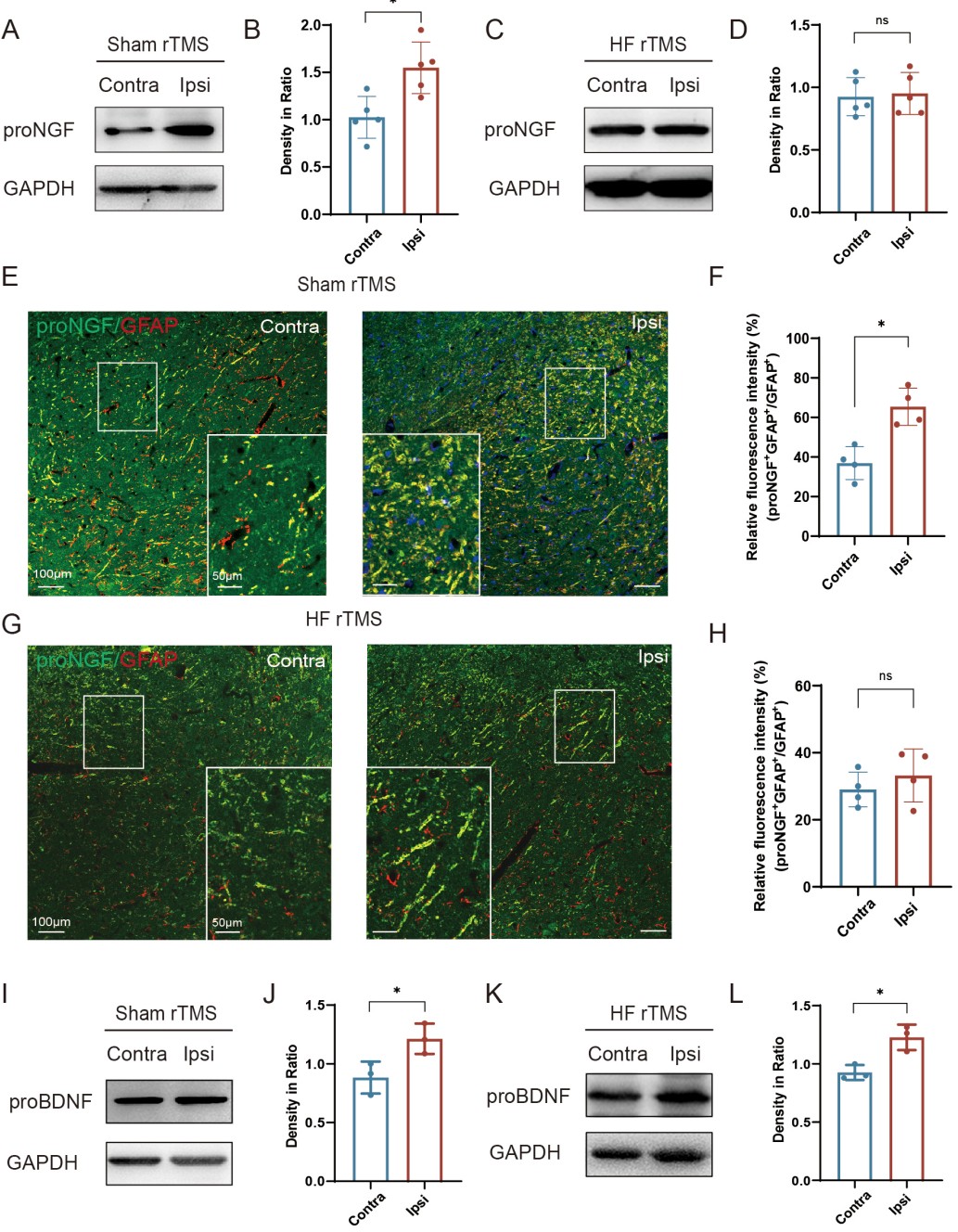

**Figure 4  HF rTMS reduced the expression of proNGF in astrocytes without affecting the expression of proBDNF.** (A) Representative Western blot of proNGF level in the ipsilateral or contralateral ventral midbrain in sham rTMS. (B) Quantitative summary levels of proNGF expression. (C) Representative Western blot of proNGF level in the ipsilateral or contralateral ventral midbrain in HF rTMS. (D) Quantitative summary levels of proNGF expression. (E) Representative immunofluorescence demonstrating GFAP and proNGF positive neurons in the contralateral or ipsilateral SNpc in sham rTMS. (continued on next page...)

**Figure 4 (…continued)**
(F) Relative fluorescence intensity (%) of proNGF + GFAP +/GFAP + in the contralateral or ipsilateral SNpc in sham rTMS. (G) Representative immunofluorescence demonstrating GFAP and proNGF positive neurons in the contralateral or ipsilateral SNpc in HF rTMS. (H) Relative fluorescence intensity (%) of proNGF + GFAP +/GFAP + in the contralateral or ipsilateral SNpc in HF rTMS. (I) Representative western blot of proBDNF level in the ipsilateral or contralateral ventral midbrain in sham rTMS. (J) Quantitative summary levels of proBDNF expression. (K) Representative western blot of proBDNF level in the ipsilateral or contralateral ventral midbrain in HF rTMS. (L) Quantitative summary levels of proBDNF expression. Data are expressed as mean ± SD ($n = 5$/group or $n = 3$/group) (*$P < 0.05$ by paired $t$-test).

frequency. In addition to frequency, duration is an essential factor affecting rTMS treatment efficacy. Our previous study reported that two weeks of HF or LF rTMS is insufficient to improve motor behavior (*Kang et al., 2022*). Present and previous studies demonstrated that a four-week regimen of HF rTMS significantly ameliorates APO-induced rotational impairment. Another previous study reported that after six weeks of rTMS treatment, there is a reduction in inflammasomes and pyroptosis-related effector molecules, a decrease in DA neuronal damage, and an enhancement in swallowing quality in MPTP-induced PD mice, exhibiting a significant improvement compared to 4 w of rTMS treatment (*Huang et al., 2024*). These findings of basic studies suggest that prolonged rTMS treatment may yield a more significant effect. This may explain the inability to identify the beneficial effects of rTMS (administered over 4 d) (*Filipovic, Rothwell & Bhatia, 2010*) or intermittent theta-burst stimulation, a new excitatory rTMS (administered for 2 w) (*Benninger et al., 2011*) in previously discussed clinical studies. Future clinical studies should assess the treatment duration to comprehensively clarify the full efficacy of rTMS in PD. In our study, we observed that the proNGF-p75NTR-sortilin complex plays a significant role in the degeneration of nigrostriatal DA neurons and the progression of PD. This finding is consistent with Professor Chen Liangwei 's report that the presence of the proNGF-sortilin signaling complex in nigral DA neurons following 6-OHDA injury may lead to neuronal apoptosis or neurodegeneration in the pathogenesis and disease progression of PD (*Chen et al., 2008*). This consistency suggests that further investigation of these underlying mechanisms can facilitate the development of new neuroprotective strategies targeting the proNGF-p75NTR-sortilin signaling cascade for PD treatment. However, our results also revealed that HF rTMS mitigated the abnormal upregulation of the proNGF-p75NTR-sortilin complex in the ventral midbrain and reduced the apoptosis of nigrostriatal dopamine neurons in the midbrain after PD. This indicates that understanding how HF rTMS influences this signaling cascade could provide valuable insights into its therapeutic mechanism and pave the way for novel treatments targeting this pathway in PD. Our study demonstrated that in the pathological progression of PD, there is an increase in proNGF expression in reactive astrocytes, and p75NTR expression and sortilin in neurons. These findings suggest that the proNGF-p75NTR-sortilin pathway is involved in the neurodegenerative process of PD, rather than merely reflecting the injury associated with 6-OHDA injection. These factors collectively facilitate proNGF-p75NTR-sortilin complex formation, resulting in neuronal death. Under pathological and immune stimulation in PD, astrocytes undergo a transition into a reactive state characterized by increased release

of inflammatory and cytotoxic mediators (*Kang et al., 2022*; *Ding et al., 2021*; *Lee et al., 2023*; *Kam et al., 2020*). HF rTMS administered over 4 w inhibited astrocyte activation and downregulated proinflammatory factors, including tumor necrosis factor α (TNF-α), interleukin-1β (IL-1β), and IL-6 in 6-OHDA rat PD model and C6 astroglial cells (*Kang et al., 2022*). A previous study reported that HF rTMS targeting the left insular, anterior dorsal granular region regulated proinflammatory factor expression, resulting in lower levels than those observed in the chronic constriction injury group (*Hu et al., 2022*). Additionally, they confirmed that rTMS can modulate the inflammatory signaling pathways associated with mGluR5/NMDAR2B (*Hu et al., 2022*). In addition to proinflammatory factors, the expression and secretion of proNGF by astrocytes have been demonstrated by numerous experiments (*Domeniconi, Hempstead & Chao, 2007*; *Domowicz et al., 2008*; *Volosin et al., 2008*; *Volosin et al., 2006*). Activated astrocytes secrete toxic molecules of proNGF-promoting motor neuron cell death (*Domeniconi, Hempstead & Chao, 2007*). *Volosin et al. (2008)* and *Volosin et al. (2006)* reported that proNGF levels increased in hippocampal and basal forebrain astrocytes of rats after seizures induced by pilocarpine and kainic acid. This is the first study to demonstrate that HF rTMS decreases the proNGF expression in astrocytes, accompanied by the inhibition of astrocytic activation. Furthermore, HF rTMS can reduce p75NTR expression and sortilin in neurons, providing an additional protective role. Previous study demonstrated proNGF, sortilin, and p75NTR expression in the dorsal root ganglion (DRG), with a subset of neurons coexpressing sortilin and p75NTR (*Arnett, Ryals & Wright, 2007*). Within 25 days of nerve injury, a significant decrease in the number of small-diameter DRG neurons that coexpress sortilin and p75NTR was observed (*Arnett, Ryals & Wright, 2007*). Furthermore, IL-1β can enhance neuronal sensitivity to proNGF by facilitating the surface accumulation of p75NTR and its accessory receptor sortilin, consequently inducing neuronal death (*Choi & Friedman, 2014*). These findings highlight the crucial role of sortilin and p75NTR in proNGF-induced neuronal death (*Arnett, Ryals & Wright, 2007*; *Choi & Friedman, 2014*; *Nykjaer et al., 2004*). Consequently, inhibiting sortilin and p75NTR expression may mitigate neuronal sensitivity to proNGF, enhancing their survival rate.

Similar to proNGF, the BDNF precursor proBDNF induces mitochondrial apoptosis in satellite glial cells and neurons by proBDNF-p75NTR-sortilin signaling pathways (*Ali et al., 2024*). Previous study demonstrated that HF rTMS upregulation of BDNF significantly improved spatial cognition and hippocampal synaptic plasticity impairments in PD (*Kaminska et al., 2022*). Previous studies have reported that neurons lacking p75NTR are resistant to proBDNF-induced apoptosis, and a competitive antagonist of sortilin can prevent sympathetic neuron death. Our study demonstrated that HF rTMS did not significantly affect proBDNF expression, indicating that HF rTMS is selective for proNGF. In our study, we observed that HF rTMS did not significantly affect the expression of proBDNF, which was somewhat unexpected. This suggests that the selectivity of HF rTMS may be due to the distinct signaling pathways and cellular effects associated with proNGF and proBDNF. The regulatory mechanisms of proBDNF may differ from those influenced by HF rTMS. This finding highlights the complexity of neurotrophic factor regulation and underscores the need for further investigation into the expression mechanisms of proBDNF

in the context of PD and related interventions. The mechanism of this phenomenon deserves further investigation.

This study has some limitations. First, the rTMS coils used were commercially available and designed for rodents, although they remain relatively large for the rat brain. Future studies should develop more precise coils tailored to the size of the rat brain. Second, rat head movement was manually restrained and visually monitored during stimulation, which could not remain completely stationary, potentially affecting the transcranial magnetic stimulation efficacy.

In addition to the currently discovered proNGF mechanism, the effects of rTMS may involve multiple mechanisms. In recent years, studies have explored the modulation of dopamine receptors by rTMS. For example, one study used high-affinity dopamine D2 receptor radioligand imaging technology and found that HF rTMS (10 Hz) stimulation of the left dorsolateral prefrontal cortex (DLPFC) significantly increased dopamine release in the medial prefrontal cortex (including the anterior cingulate cortex and orbitofrontal cortex) (*Etievant et al., 2015*). This suggests that rTMS may exert its therapeutic effects through the modulation of the dopamine system. Such modulation may be achieved by enhancing the excitatory effects mediated by D1 receptors or inhibiting the inhibitory effects mediated by D2 receptors. However, the current studies have not yet clarified the direct mechanisms of rTMS on dopamine receptors, especially the specific effects under different brain regions and stimulation parameters.

Furthermore, the therapeutic effects of rTMS may also depend on specific subtypes of dopamine receptors. For example, D1 receptors play a key role in enhancing long-term potentiation (LTP)-like plasticity, while D2 receptors mainly regulate neural activity by inhibiting glutamatergic transmission (*Darvish-Ghane et al., 2020*). Therefore, future studies need to further explore the direct mechanisms of rTMS on dopamine receptors and how to optimize the therapeutic effects of rTMS by modulating these receptors.

In conclusion, this study demonstrated the neuroprotective potential of HF rTMS in reducing PD neuron apoptosis post-PD. Further investigation of these underlying mechanisms can facilitate the development of new neuroprotective strategies targeting the proNGF-p75NTR-sortilin signaling cascade for PD treatment.

## CONCLUSION

The proNGF-p75NTR-sortilin complex plays a significant role in the degeneration of nigrostriatal DA neurons and PD progression. HF rTMS mitigated the abnormal upregulation of the proNGF-p75NTR-sortilin complex in the ventral midbrain following 6-OHDA injury and reduced the apoptosis of nigrostriatal dopamine neurons in the midbrain after PD (Fig. 5).

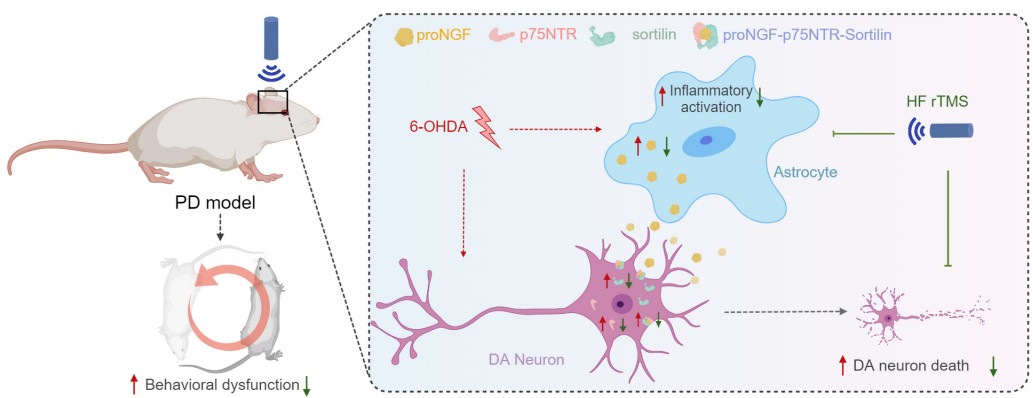

**Figure 5** **HF rTMS protects against 6-OHDA-induced PD symptoms by modulating the proNGF-p75NTR-sortilin pathway.** Herein, a significant degree of apoptosis and pronounced rotational behavior was observed in ipsilateral DA neurons following 6-OHDA-induced injury. However, the survival of these DA neurons was significantly enhanced subsequent to HF rTMS treatment. This study demonstrated that HF rTMS has a significant anti-inflammatory effect. By inhibiting astrocyte activation and selectively reducing their proNGF expression, it decreases the surge in proNGF-p75NTR-sortilin complexes. HF rTMS alleviates the death loss of DA neurons by reducing these complexes, providing a potentially effective therapeutic strategy for PD treatment.

## Funding

This work was funded by grants from the National Natural Science Foundation of China (Nos. 81802229, 82272591). The funders had no role in study design, data collection and analysis, decision to publish, or preparation of the manuscript.

## Grant Disclosures

The following grant information was disclosed by the authors:
National Natural Science Foundation of China: Nos. 81802229, 82272591.

## Competing Interests

The authors declare there are no competing interests.

## Author Contributions

- Rui Zhao conceived and designed the experiments, performed the experiments, analyzed the data, prepared figures and/or tables, and approved the final draft.
- Wanqing Du performed the experiments, analyzed the data, prepared figures and/or tables, and approved the final draft.
- Ke Tian performed the experiments, analyzed the data, prepared figures and/or tables, and approved the final draft.
- Kunlong Zhang conceived and designed the experiments, authored or reviewed drafts of the article, and approved the final draft.
- Hua Yuan conceived and designed the experiments, authored or reviewed drafts of the article, and approved the final draft.

- Fang Gao conceived and designed the experiments, authored or reviewed drafts of the article, and approved the final draft.
- Xin Kang conceived and designed the experiments, authored or reviewed drafts of the article, and approved the final draft.
- Xiaolong Sun conceived and designed the experiments, authored or reviewed drafts of the article, and approved the final draft.

## Animal Ethics

The following information was supplied relating to ethical approvals (i.e., approving body and any reference numbers):

The Committee of Animal Use for Research and Education of FMMU provided full approval for this research (IACUC-20230558).

## Data Availability

Raw data is available in the Supplemental Files.

## Supplemental Information

Supplemental information for this article can be found online at http://dx.doi.org/10.7717/peerj.19633#supplemental-information.

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
