# Peer review of "High-frequency repetitive transcranial magnetic stimulation protects against 6-OHDA-induced Parkinson’s disease symptoms by modulating the proNGF-p75NTR-sortilin pathway"

_PeerJ, doi:10.7717/peerj.19633_

## Round 0.1 · original submission · Major Revisions

Thank you for submitting your manuscript. We have carefully reviewed your work and appreciate the insights it offers to the field. While your submission has great potential, the reviewers have raised some valid concerns that need to be addressed for the manuscript to reach its full potential. Kindly take the time to thoroughly address each of the raised issues, providing clear justifications for the choices you make.

Reviewer 1 ·

Basic reporting

The overall basic reporting of this article was good, however, there are several areas of improvement I will suggest to enhance understanding, starting with the introduction:
1. The introduction needs some more background on the proNGF-p75NTR-sortilin pathway in relation to Parkinson's Disease. This remains an issue in the conclusion where the authors cite studies on nerve injury relating to this pathway, but not about PD specifically (lines 300-304). This emphasis on nerve injury makes the reader wonder if what was really being assessed was how proNGF, sortilin, and p75NTR are involved in the injury that accompanies 6-OHDA injection, and not the effects of PD itself.
2. Similarly, in lines 70-72, the authors discuss the therapeutic effects of rTMS in humans, but do not specifically cite clinical articles on PD. It would be helpful to cite evidence that is specific to the disorder that is being investigated.
3. I think the authors could be more specific about the side effects of PD medications and other treatments in lines 64-66.
4. The lines 226-227 would have been helpful to include in the introduction since it was unclear what sortilin’s function is until this point.

Additionally, throughout the article, there are minor grammatical issues that could be remedied with additional editing.

The figures and their labels are well done!

Experimental design

The experimental question was well-defined, however, there are several issues relating to the design of the experiments performed which make this section difficult to understand.

1. The specific cohorts of rats used were not described in the methods section, which I found confusing. In lines 101-102, the authors state that rats were euthanized using CO2, but in line 151, they state that the rats were euthanized by decapitation. Then, in lines 172-174, the authors state that the animals were euthanized by transcardial perfusion. I am assuming these are separate cohorts of animals, but the number of animals per group and per cohort is not specified, making this section unclear. The groups of animals used for each analysis should be clearly stated in the methods section, not just in the captions of figures.
2. Because of the above issue, it is hard to comment on the statistical analyses that were used in that section. However, in line 196, the authors state that post-hoc tests were either Tukey’s, Dunnett’s or Bonferroni. It should be specified which one of these post-hoc tests were done per analysis in the figures or body of the article.
3. It is also unclear in the section about the immunohistochemical staining on lines 184-189 if the antigens labeled with the anti-mouse secondary green antibody were all labeled in the same tissue sections. I do not believe the authors did this, as it would stain multiple targets green that were stained with a mouse primary antibody, but the way this is written leads the reader to believe that all tissues were stained with all antibodies.
4. I think the schematic of the experimental design in Figure 1A was very helpful for understanding the general experimental design! However, more explanation in the methods section above is necessary.
5. The pro-BDNF staining mentioned in line 246 was not mentioned in the Methods section under “Immunofluorescent labeling.”

Validity of the findings

For the findings, I believe there needs to be some additional data shown and some more specific language used to describe these findings as to not be misleading. Specifically,

1. The immunofluorescence images in Figures 2, 3, and 4 are not accompanied by quantification of the staining so the potential significance of this analysis is not clear. I assume this was done by quantification of intensity of signal in the tissue sections so those results should be in the figures as well. For example, in lines 220-221, the authors stated that they saw a “significant decrease in ipsilateral sortilin fluorescence intensity” but there is no compiled data of intensity measurements across subjects’ tissues in the figure to support this statement.
2. The statement above, on lines 219-220, states that there was a significant increase in morphology associated with sortilin expression. This statement does not make sense, as the authors never mentioned doing a morphology analysis in the methods section. Do the authors mean that there is a morphology change? Or that there is simply more sortilin expression? The wording needs to be more clear here.

In the Discussion/Conclusion section, there are some specific areas of improvement that I think would make the take-away message from this article more clear.

1. In lines 283-297, the authors spend a lot of time summarizing previous articles without tying them to the current study. I think the manuscript could be improved by interweaving the previous findings with the current study instead of large blocks of just discussing other articles.
2. In the first line of the conclusion (line 328), the authors state that the “proNGF-p75NTR-sortilin complex is essential in the degeneration of nigrostriatal dopamine neurons and PD progression.” The use of the term “essential” is an overstatement and not a claim that can be made based on the experiments performed. The experiments showed that hf-rTMS has decreased these complexes, but not that they are necessary for hf-rTMS to work.
3. Figure 5 illustrates the theory behind the experiments but is a little confusing. For example, it looks as though 6-OHDA is only affecting the astrocytes and not the neurons themselves. I do still believe that a figure illustrating the hypothesis is a good idea, but this figure may be more informative if it is simplified.
4. In lines 315-317, it is stated that proBDNF was not affected but the authors make no comment on whether or not this was what they were expecting and do not explain what could be contributing to this finding.
5. In lines 319-320, the authors state that a future consideration is to develop smaller TMS coils, however, due to the physical constraints, this may not be possible. There are other limitations, such as small sample size, that could be discussed.

·

Basic reporting

This is an interesting article titled “High-frequency repetitive transcranial magnetic stimulation protects against 6-OHDA-induced Parkinson's disease symptoms by modulating the proNGF-p75NTR-sortilin pathway”.
Major points:
 Studying the dopamine receptors in the future might give an idea about the type of dopamine receptor involved. Authors can provide this as limitations/ Future Directions.
 Whether HF rTMS influences the D1 or D2 family, studying the D1 and D2 receptor gives an idea about the excitatory and inhibitory effects of HF rTMS on PD, authors need to clarify
Minor points:
 Line 37- Metheds spelling
 Line 64, 65- pharmacological and DBS side effects not stated and explained well
 Line 260- Mentioning about the autonomic nervous system and HF rTMS may not be relevant. If not, please state the relation
 Line 296- Mentioning about spinal cord astrocytes neonatal mice may not be relevant, if not please state the relation.
 Only male rats – included in the study
 There is no inclusion of a Control group for motor task comparison
 Figure 1(D)- One outlier in the graph driving that into significance. Authors need to mention that
 Line 233- space and coma
 Line 139-rTMS- site of stimulation not specified

Experimental design

Good study design

Validity of the findings

Looks valid

Additional comments

If authors modify as per the following suggestions, the modified manuscript can be considered for publication at the Editor’s discretion.

---

## Round 0.2 · accepted · Accept

Reviewer 1 and myself assessed the revision and confirm that he authors have addressed all of the reviewers' comments. The manuscript is now ready for publication.

Reviewer 1 ·

Basic reporting

The edits that the authors made helped in giving more background to the article. I appreciate the work they put in to give a more comprehensive explanation about this pathway in the introduction.

Experimental design

I thank the authors for clarifying parts of their experimental design that were previously unclear. The quantifications of the staining were especially helpful!

Validity of the findings

No comment